# Detection and Prevalence of Macrolide and Fluoroquinolone Resistance in *Mycoplasma genitalium* in Badalona, Spain

**DOI:** 10.3390/antibiotics11040485

**Published:** 2022-04-05

**Authors:** Belén Rivaya, Chloé Le Roy, Elena Jordana-Lluch, Gema Fernández-Rivas, Cristina Casañ, Victoria González, Jun Hao Wang-Wang, Cécile Bébéar, Lurdes Matas, Sabine Pereyre

**Affiliations:** 1Microbiology Department, Laboratori Clinic Metropolitana Nord, Hospital Universitari Germans Trias i Pujol, Ctra. Del Canyet, S/N, 08916 Badalona, Spain; belen.rivaya@sespa.es (B.R.); elena.jordana@ssib.es (E.J.-L.); ccasan.germanstrias@gencat.cat (C.C.); vgsoler@iconcologia.net (V.G.); jhwang.germanstrias@gencat.cat (J.H.W.-W.); lurdesmatasa@gmail.com (L.M.); 2Department of Genetics and Microbiology, Universitat Autònoma de Barcelona, 08193 Cerdanyola del Vallès, Spain; 3UMR 5234 Fundamental Microbiology and Pathogenicity, University of Bordeaux, CNRS, F-33000 Bordeaux, France; chloe.le-roy@u-bordeaux.fr (C.L.R.); cecile.bebear@u-bordeaux.fr (C.B.); sabine.pereyre@u-bordeaux.fr (S.P.); 4Institute for Health Science Research Germans Trias i Pujol (IGTP), Ctra. Del Canyet, S/N, 08916 Badalona, Spain; 5Center for Epidemiological Studies on HIV/AIDS and STI of Catalonia (CEEISCAT), Generalitat de Catalunya, Ctra, Del Canyet, S/N, 08916 Badalona, Spain; 6CIBER in Epidemiology and Public Health (CIBERESP), Avda. Monforte de Lemos, 3-5, 28029 Madrid, Spain; 7Bacteriology Department, National Reference Centre for Bacterial Sexually Transmitted Infections, CHU Bordeaux, F-33000 Bordeaux, France

**Keywords:** *Mycoplasma genitalium*, macrolide, fluoroquinolone, resistance, Spain

## Abstract

Macrolide and fluoroquinolone resistance (MLr/FQr) in *Mycoplasma genitalium* (MG) infections is concerning worldwide. Current guidelines recommend performing MLr detection in MG-positive cases to adjust antimicrobial therapy. We aimed to evaluate the usefulness of PCR followed by pyrosequencing for MLr detection in comparison with a one-step commercial assay and to assess the prevalence of MLr and FQr in Badalona, Spain. A total of 415 MG-positive samples by Allplex STI-7 (Seegene) were analyzed for MLr detection by pyrosequencing. From those, 179 samples were further analyzed for MG and MLr by ResistancePlus^®^ MG kit (SpeeDx) and 100 of them also for fluoroquinolone resistance (FQr) by sequencing the *parC* gene. Regarding MG detection, Allplex and Resistance Plus^®^ showed an overall agreement of 87%, but this value rose to 95.4% if we compare them for MLr detection. Prevalence of MLr was 23.1% in Badalona, but this rate increased to 73.7% in the HIV-positive patients cohort. FQr detection showed 3% of resistant strains. Pyrosequencing is a convenient and cheap technique for MLr detection, but one-step tools should be considered in high-throughput laboratories. Despite the fact that MLr remained moderate and FQr was low in our study, simultaneous MG and MLr detection would improve patient’s management applying resistance-guided treatment strategies.

## 1. Introduction

*Mycoplasma genitalium* (MG) is an important sexually transmitted pathogen responsible for 15–20% of nongonococcal urethritis (NGU) in men [1] and cervicitis, pelvic inflammatory disease, preterm birth, and abortion in women [2]. MG is a fastidious microorganism that is difficult to grow from clinical samples and cellular culture is generally limited to reference laboratories [3]. For this reason, molecular detection is the most widely used method for routine diagnosis of MG infection.

MG lacks a cell wall, being intrinsically resistant to beta-lactam antibiotics. Macrolides (ML) have been extensively used worldwide as a first-line treatment, with quinolones as second-line therapy. However, given the high capacity of this microorganism for acquiring antibiotic resistance to both groups, MG becoming an untreatable sexually transmitted infection (STI) is a plausible and close scenario. European guidelines recommend an extended ML regimen to avoid macrolide resistance (MLr) acquisition in susceptible strains [4]. The fourth-generation fluoroquinolone moxifloxacin is recommended as a second option if the strain is resistant to macrolides [4]. Furthermore, treatment failures with both therapies have already been reported—given by dual-resistant strains [5,6]. Alternatively, doxycycline has also been used but with poor efficacy [7]. For this reason, pristinamycin has been established as the primary third-line treatment for patients with ML or fluoroquinolone treatment failure [4].

Therefore, according to current guidelines in STI, MG detection in symptomatic patients and subsequent MLr studies on positive cases should be performed to adjust antimicrobial therapy when ML resistance patterns are detected [4].

Macrolide resistance is caused by point mutations in region V of the 23S rRNA gene at positions 2058, 2059 or 2062 (*E. coli* numbering). These mutations can be detected by in-house techniques using a two-step approach performing MG detection first and analyzing afterwards MLr presence in MG-positive cases [8,9]. Recently available commercial kits can also be used [10], with many of them able to detect MG and MLr patterns in a single step. Similarly, fluoroquinolone resistance (FQr) is mainly caused by mutations in genes *parC* or *gyrA* [11] and detecting these patterns when MLr is detected or especially after moxifloxacin treatment failure could be an interesting option to reserve third-line antimicrobials for selected patients. Quinolone resistance-determining regions (QRDR) can be studied by Sanger sequencing [12] or by new commercial assays authorized only for research purposes [13,14].

Limited data are available regarding MG macrolide and fluoroquinolone resistance in Spain [15,16,17,18]. The aim of this study was to evaluate pyrosequencing as an adequate in-house technique for detecting MG MLr in our two-step routine, by comparing it to a commercial one-step assay and to assess the prevalence of macrolide and fluoroquinolone resistance in Badalona, Spain.

## 2. Results

### 2.1. Demographic Data and Clinical Findings

A total of 415 MG-positive specimens detected by Allplex STI-7 assay (Seegene Inc., Seoul, Korea) from 407 individuals were collected between 2016 and 2018 from people who required STI diagnosis, being females in 75.9% of cases (*n* = 309) and males in 24.1% (*n* = 98). The mean age of the studied population was 26.2 years (ranging from 16 to 68).

The highest number of positive results was detected in patients up to 25 years-old (*n* = 199, 47.95%), followed by patients from 26 to 40 years-old (*n* = 158, 38.07%) and older than 40 years (*n* = 58, 13.98%). More details about the samples analyzed and their origins can be found in Table 1.

Information regarding clinical findings was registered by clinicians during the prospective part of study. Epidemiological information, medical records about symptomatology, or reason for seeking a STI test was obtained for 136 out of 181 patients included, showing that the most common cause was a previous intercourse with a STI-positive patient, followed by urethritis and leukorrhea, as shown in Figure 1. More detailed information about epidemiological data and clinical findings is presented in Table 2.

Focusing on risk factors for a MG infection, our cohort included 38 known HIV-positive males, including 94.7% (*n* = 36) men who have sex with men (MSM). Most of them were asymptomatic (63.2%, *n* = 24) and, among those with symptoms, proctitis (15.8%, *n* = 6), urethritis (13.2%, *n* = 5), or dysuria (7.9%, *n* = 3) were the most prevalent. Remarkably, in the HIV-positive group, MG was the only pathogen detected in almost half of the cases (*n* = 17, 44.7%).

Coinfections with other STI agents were detected, such as *Chlamydia trachomatis* (*n* = 1, 2.6%), *Neisseria gonorrhoeae* (*n* = 3, 7.9%), and *Treponema pallidum* (*n* = 6, 15.8%, one case jointly with *N. gonorrhoeae*). Other microorganisms such as *Ureaplasma* spp. or *M. hominis* were found in 31.6% of patients but were not considered because of their uncertain pathogenic role.

### 2.2. Detection of Macrolide and Fluoroquinolone Resistance

Among the 415 MG-positive samples detected by Allplex STI-7, 21 contained insufficient DNA to perform the resistance study, leaving a total of 394 specimens. Macrolide resistance analysis was performed by two different methods: pyrosequencing, ResistancePlus^®^ MG kit, or by both methods. A subset of 100 samples was analyzed by both methods, whereas additional 215 samples were processed by pyrosequencing only and 79 more by ResistancePlus^®^ MG kit only. The prevalence of macrolide resistance was 23.1%, with a total of 91 samples harboring macrolide resistance-associated mutations. More details are shown in Figure 2.

Macrolide resistance prevalence was also calculated in both study periods, observing a slight increase in 2018 (26%) in relation with the previous phase (22.2%).

Interestingly, when focusing on the HIV cohort only, the prevalence of macrolide resistance was 73.7% (28/38), with the clinical records showing a previous antibiotic regimen in 19.8% of cases, predominantly macrolides.

A total of 72 specimens had a mutation conferring MLr detected by pyrosequencing and this method revealed four different types of mutation. The most prevalent mutation was A2058G (*n* = 41, 56.9%), followed by A2059G (*n* = 27, 37.5%), A2058T (*n* = 3, 4.2%), and A2062T (*n* = 1, 1.4%).

The subset of 100 samples with MLr detected by pyrosequencing was further analyzed for MLr by the ResistancePlus^®^ MG kit and for fluoroquinolone resistance detection. As the ResistancePlus^®^ MG assay offers MG detection data, a comparison between that assay and the Allplex STI-7 assay could be performed, with an overall agreement regarding MG detection of 87% between both techniques. When MLr detection was compared, the agreement between the ResistancePlus^®^ MG kit and pyrosequencing was 95.4%.

Regarding fluoroquinolone resistance (FQr), 97 out of the 100 samples showed a wild-type *parC* gene, whereas three of them had the mutation Ser83(80)Ile. Remarkably, these three strains were also macrolide resistant, corresponding in two cases to men with urethritis and one 18-year-old woman for whom the first sample (collected after an empiric 5 day-course ML treatment) showed an A2059G mutation/*parC* wild-type strain. The woman was subsequently treated with moxifloxacin and the test of cure showed a double resistance pattern (A2059G/Ser83(80)Ile) after fluoroquinolone therapy.

## 3. Discussion

*Mycoplasma genitalium* is a pathogen associated with NGU, balanoposthitis, and chronic prostatitis in men, as well as cervicitis and PID among other consequences in women [1]. Treatment failures are especially concerning due to MG’s capacity of acquiring mutations that confer macrolide or fluoroquinolone resistance. Since 1983, when MG was described as a new species in the urogenital tract [19], multiple reports focusing on antibiotic resistance have been published in Europe [20] and worldwide [21]. Unfortunately, there are limited data of MG infections and resistance patterns in Spain [15,16,17,18].

Most of the MG-positive results came from people below 25 years-old in our study, related to the epidemiological surveillance system to monitor *C. trachomatis* (CT) performed in Catalonia since 2007. This fact implies that the higher burden of samples received in the microbiology departments for STI testing belong to this population group. Among the HIV cohort, rectal positivity was higher in asymptomatic patients, as previously described by Read et al. [22].

In our setting, multiplex PCR detecting seven STIs and urogenital bacteria (*C. trachomatis*, *N. gonorrhoeae*, *M. genitalium*, *T. vaginalis*, *M. hominis*, *U. urealyticum*, and *U. parvum*) is currently employed [23]. However, according to most international guidelines [4,24], MG testing is only recommended based on presence of symptoms or in current sexual partners of persons infected with MG in order to avoid unnecessary treatments leading to antibiotic resistance [25,26]. In our case, if MG is detected, clinical records are checked to look for symptoms. If there is no history of any patient’s symptomatology or if the patient is not a sexual partner of a positive case, MG positivity is not displayed. In the same way, commensal bacteria such as *Ureaplasma* spp. or *M. hominis* can be detected with the multiplex PCR simultaneously to “true” STI agents. Given their uncertain pathogenic role and controversial treatment benefits [27], these bacteria are only reported in exceptional cases to avoid antimicrobial resistance development.

A comparison between the Allplex STI-7 and the ResistancePlus^®^ MG kit for the detection of MG was performed in a subset of 100 samples, with an overall agreement of 87% (87/100). For 13 discordant cases, ResistancePlus^®^ MG kit did not detect MG while Allplex gave a positive detection, with a subsequent successful 23S rRNA gene pyrosequencing. Therefore, the MG-negative results by the ResistancePlus^®^ MG kit could be explained by low bacterial load plus potentially loss of DNA integrity due to an additional freeze–thaw cycle (specimens were first tested by the Allplex STI-7 assay and then frozen at −20 °C until tested by ResistancePlus^®^ MG). The target used for MG detection by Allplex STI-7 is unknown, while the assay ResistancePlus^®^ MG amplifies the MgPa gene. Thus, a higher sensitivity in the target used by Allplex could also be responsible for its superior performance.

Pyrosequencing was employed as our main technique to detect MLr, given our prior experience of MLr mutation detection in *M. pneumoniae* [28]. Samples from quality controls were used as references, demonstrating the robustness of the developed method. The commercial assay ResistancePlus^®^ MG kit was also evaluated in a subset of 100 samples. This assay is very convenient for high-throughput laboratories and can be easily implemented in routine diagnosis, allowing MG detection and macrolide resistance determination as a one-step method without requiring specialized instrumentation and covering that need reflected in the current guidelines [4]. However, this assay is not suitable for epidemiological or research purposes on mutation types as it only detects a specific set of 23S rRNA variations associated with MLr but does not distinguish among them nor detects new ones, in contrast to pyrosequencing. Additionally, the lack of a 23S internal control for MLr targets implies that this kit does not discern between absence of mutation and absence of 23S rRNA gene amplification, leading to a risk of false susceptible results when the 23S rRNA gene is not amplified, as previously reported by Le Roy et al. [29]. When compared with pyrosequencing, the overall agreement was 95.4% (83/87). Among the four discordant results, one mutated strain by pyrosequencing was informed as negative by ResistancePlus^®^ MG due to late detection (Ct > 30), which might correspond with low bacterial load. Unfortunately, due to insufficient material, those four specimens could not be tested again.

Overall, the macrolide resistance rate detected in our setting was 23.1%. Our rate is similar to other studies performed in different parts of our country [30,31,32]—slightly higher than the 16.3% detected in the Basque Country [17] but lower than the 36.4% reported by de Salazar et al. [33] in southern Spain. However, their cohort included 25% of MSM versus 9.1% in our cohort. Focusing on Catalonia, our prevalence of resistance is similar to the one reported by Muñoz-Santa et al. (23%) [34] and between the rates published by Lucena-Nemirosky et al. (12.6%) in a routine diagnostic service [18] and Fernández-Huerta et al. (36.1%) in a STI Unit [15].

Among the HIV cohort, given that 94.7% of them were MSM, the high rate of MLr (73.7%) is consistent with previous data reported by Barberá et al. in Spain [35], demonstrating that MSMs have a higher risk of acquisition of macrolide-resistant MG, and by Dionne-Odom et al.’s study, reporting a prevalence of multidrug resistant MG of 74.1% in HIV-MSM patients in Alabama [36]. The subsequent risk of treatment failure highlights the need to perform diagnosis and macrolide resistance detection simultaneously.

Fluoroquinolone resistance (FQr) detection performed in a subset of samples (*n* = 100) showed 3% of resistant strains that harbored Ser83Ile mutation in the *parC* gene. This rate is slightly lower than those described in our country by Fernández-Huerta et al. (8.8%), Piñeiro et al. (7.9%, but not all of them were clearly implicated in treatment failure), de Salazar et al. (9.1%), and Adelantado and Beristain (5.5%) [13,17,32,33]. However, our results are concordant with the lower FQr prevalence detected in WHO European regions, in comparison with Western Pacific territories [21].

This study has several limitations. Firstly, our study population comprehended both symptomatic and asymptomatic individuals. This fact could affect macrolide and fluoroquinolone resistance values, in comparison with other Spanish studies [15,33]. Secondly, it was not possible to test all samples for ML resistance with both methods simultaneously. The use of only 100 samples for the comparison of both methods is a limitation of the study but gives a good insight into the usefulness of the pyrosequencing assay. Thirdly, for fluoroquinolone resistance, only *parC* gene was sequenced for being the most frequently involved in moxifloxacin failure [11].

## 4. Materials and Methods

### 4.1. Study Setting, Patients and Sample Collection, and Study Design

The Laboratori Clínic Metropolitana Nord, Germans Trias i Pujol University Hospital (Badalona, Spain) is a laboratory with an influence area covering a total of 1,500,000 inhabitants with more than 100 primary care facilities—including Sexual Reproductive Health Centers (ASSIR)—that centralizes the analysis of all the specimens together with the ones derived from the tertiary hospital Germans Trias i Pujol.

A total of 415 samples were collected by physicians between 2016 and 2018 from 407 patients for whom STI testing was ordered, including symptomatic, asymptomatic, or STI contacts.

Most samples (*n* = 371) belonged to young people recruited in ASSIRs for epidemiological surveillance purposes to monitor *C. trachomatis* (CT) prevalence among those under 25 years of age with high-risk sexual behavior [37]. This surveillance has been performed in Catalonia since 2007. The other 44 samples were from the HIV Unit or Gynecology and Obstetrics Department of the Germans Trias i Pujol University Hospital.

The study was designed with a retrospective and a prospective part. For the first phase, a retrospective analysis included 234 MG-positive samples received between 2016 and 2017. During the second phase, a one-year prospective study was carried out from December 2017 to November 2018. A total of 181 MG-positive samples were collected from 173 patients that required STI testing at our medical institution or ASSIR Centers (Badalona and Mataró, Barcelona, Spain). Detailed information is displayed in Table 1.

### 4.2. Clinical and Microbiological Data

Demographic data (gender, age) were recorded for all MG-positive patients, and epidemiological information such as risk factors for STI (more than 2 sexual partners in the last 6 months, new sexual partner, intermittent condom use, prior STI…), previous antibiotic therapy, symptomatology, and current treatment was registered in the prospective study when available (136 out of 181 patients included in the prospective evaluation). Microbiological data such as the detection of concomitant STIs, syphilis, or HIV, was provided by the Microbiology Department. More information can be found in Table 2.

### 4.3. Diagnosis of Mycoplasma genitalium

A variety of samples including endocervical, urethral, or rectal swabs and urine samples were molecularly tested for sexual transmitted diseases using the Real-Time PCR STI detection Allplex STI-7 Assay (Seegene Inc., Seoul, Korea), following the manufacturer’s instructions. DNA was previously extracted by either the Microlab NIMBUS (Hamilton Company, Reno, NV, USA) or the STARlet IVD platforms (Seegene Inc., Seoul, Korea).

### 4.4. Macrolide and Fluoroquinolone Resistance Detection

MG-positive samples were subsequently investigated for macrolide resistance by pyrosequencing, as previously described [28], but using the primers described by Jensen et al. for both PCR and pyrosequencing [8]. The pyrosequencing protocol was optimized using a total of 6 MG-positive samples from external European and Spanish quality programs (Quality Control for Molecular Diagnostics program—QCMD and SEIMC Quality Control Program), sent to multiple laboratories as reference samples for MG detection and macrolide resistance analysis. Given that the mutations associated with macrolide resistance are located in region V of MG 23S rRNA gene, our prior pyrosequencing protocol [28] was modified to a single 45 cycles amplification with a final volume of 50 µL. As previously described by Spuesens et al. [38], this protocol includes four general steps: generating 5′-biotynilated products, processing PCR products to obtain single-stranded DNA where the sequence primer will anneal, sequencing, and analyzing the results.

With this purpose, PCR mixtures (50 µL) contained 0.4 µM of the biotinylated primers Mg 23S-1992Bio and Mg 23S-2138R [8], 0.2 mM of deoxynucleoside triphosphate set (Sigma Aldrich, San Luis, MO, USA), 1× Pfu Buffer (Promega Corporation, Madison, WI, USA), 0.02 U/µL of Pfu DNA polymerase (Promega), 26.6 µL of PCR grade water, and 15 µL of DNA template. The following cycling conditions were used: 5 min at 94 °C, followed by 45 cycles of 30 s at 94 °C, 30 s at 58 °C, and 30 s at 72 °C. A negative control was taken along in each PCR run. The presence of the 147-bp amplified fragment was detected by high-resolution capillary electrophoresis (QIAxcel^®^ Advanced System, Qiagen, Germany).

The resulting biotinylated products were immobilized to Streptavidin Sepharose™ High Performance beads (GE Healthcare) and processed to obtain high-quality single-stranded DNA using the Pyromark^™^ Vacuum Prep Workstation, according to manufacturer’s instructions. The immobilization and pyrosequencing reactions were based on the protocol of Spuesens et al. [38] with the following modifications: a total 4 µL of streptavidin beads and 26 µL of H_2_O per sample were used, and the plates were mixed for 10 min instead of 15 min [28].

To evaluate pyrosequencing as an adequate method for MLr study in MG, a subset of 100 samples previously processed by pyrosequencing and that included strains with A2058G (*n* = 36), A2059G (*n* = 25), or A2058T (*n* = 3) mutations and WT samples (*n* = 36) (including five samples from the European Quality Control—QCMD) was further analyzed by ResistancePlus^®^ MG kit (SpeeDx, Sydney, NSW, Australia). This kit offers the simultaneous detection of MG and 5 mutations at positions 2058 and 2059 in the 23S rRNA gene *(E. coli* numbering) that are associated with macrolide resistance.

Finally, fluoroquinolone resistance was detected in the same subset by sequencing the *parC* gene [39,40]. Both processes were carried out at the University of Bordeaux, France.

## 5. Conclusions

This study provides valuable information about *M. genitalium* macrolide and fluoroquinolone resistance rates in the young population with STI risk in Badalona, Spain, where there was a lack of data for comparison with other European countries. Pyrosequencing, despite being more time-consuming, is a cheap and useful tool to detect MLr, allowing epidemiological purposes and new mutations description. Given the rate of resistant strains detected in Badalona, Spain, simultaneous and accurate diagnosis of MG and its ML resistance patterns, using either in-house or commercial near-patient reagents that provide results without delay, should be implemented in clinical laboratories. These technologies could substantially improve patient’s clinical management applying resistance-guided treatment strategies, decreasing treatment failures, and lowering total costs.

## Figures and Tables

**Figure 1 antibiotics-11-00485-f001:**
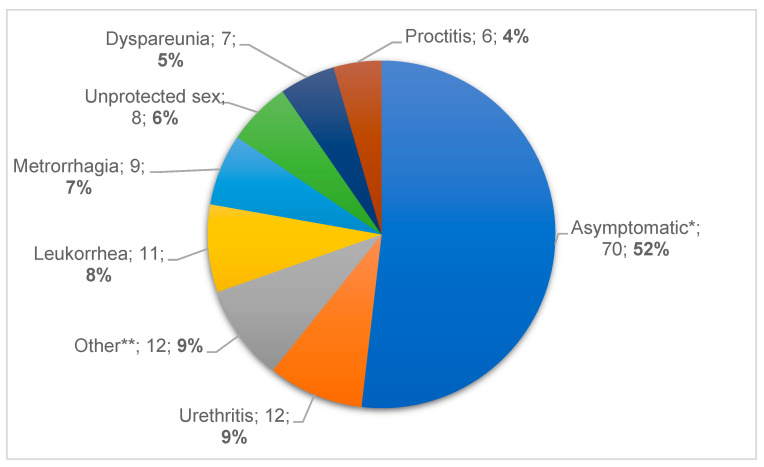
Symptoms and signs of patients included during the prospective phase of the study (Symptom, *n*, %). * Asymptomatic section includes STI contacts and patients with previous unprotected sex. ** Other: dysuria (*n* = 4), pruritus (*n* = 4), infertility (*n* = 2), test of cure (*n* = 1), amenorrhea (*n* = 1).

**Figure 2 antibiotics-11-00485-f002:**
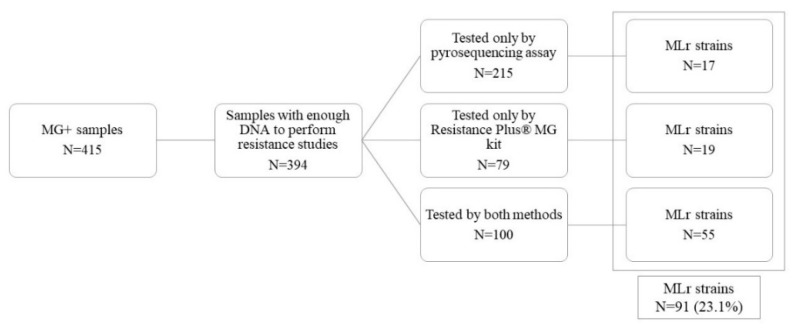
MLr strains detected during the study period.

**Table 1 antibiotics-11-00485-t001:** Samples and origin.

Origin	Sample Type				
	Endocervical Swab	Urine	Rectal Swab	Urethral Swab	Total
ASSIR *	291	73	1	6	371
HIV Unit	0	18	17	2	37
Ginecology & Obstetrics	7	0	0	0	7
Total (*n*)	298	91	18	8	415
Total (%)	71.8	21.9	4.3	1.9	100.0

* ASSIR: Sexual Reproductive Health Centers.

**Table 2 antibiotics-11-00485-t002:** Epidemiological data and clinical findings of MG infections of 136 patients included in the prospective phase of the study.

	Women (*n* = 83)	MSW (*n* = 17)	MSM (*n* = 36)
Mean age (range)	27.5 (16–56)	30.3 (20–44)	36.7 (24–55)
HIV+ (*n*, %)	-	2 (5.3%)	36 (94.7%)
Specimens (*n*, %)			
Endocervical swab	67 (80.7%)	-	-
Urine	16 (19.3%)	14 (82.3%)	17 (47.2%)
Urethral swab	-	3 (17.7%)	2 (5.6%)
Rectal swab	-	-	17 (47.2%)
Clinical findings			
Symptomatic	40 (48.2%)	8 (47.1%)	13 (36.1%)
Asymptomatic	43 (51.8%)	9 (52.9%)	23 (63.9%)
STI coinfections			
MG monoinfection	18	12	16
*Chlamydia trachomatis*	10 ^a^	-	1
*Neisseria gonorrhoeae*	-	-	3
Syphilis (data only from HIV+ patients)	-	1	4
Commensal mycoplasmas *	55	4 ^b^	12
Macrolide resistance	25/80 ** (31.2%)	11/17 (64.7%)	25/36 (69.4%)
A2058G (pyrosequencing)	14 (32.5%)	2 (11.8%)	11 (30.6%)
A2059G (pyrosequencing)	9 (20.9%)	8 (47.1%)	4 (11.1%)
A2058T (pyrosequencing)	2 (4.6%)	1 (5.9%)	-
Resistance detected (ResistancePlus^®^)	Not determined	Not determined	10 (27.8%)
No resistance detected (ResistancePlus^®^)	Not determined	Not determined	5 (13.9%)
Wild-type (pyrosequencing)	55 (35%)	6 (35.2%)	5 (13.9%)
Not performed	3 (7%)	-	1 (2.7%)
Fluoroquinolone resistance	1/28	1/9	-
Yes (*parC* mutation)	1	1	-
No	27	8	-
Not performed	55	8	-

MSW—men who have sex with women; MSM—men who have sex with men. * Commensal mycoplasmas—*M. hominis*, *U. parvum*, *U. urealyticum*. ** Three women samples could not be analyzed for MLr. ^a^ In 8/10 cases, codetection with other agents (*M. hominis*, *Ureaplasma* spp.). ^b^ In all cases, codetection with *U. parvum*.

## Data Availability

Data are available on reasonable request from the corresponding author.

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
