# Peer review of "Detection and Prevalence of Macrolide and Fluoroquinolone Resistance in Mycoplasma genitalium in Badalona, Spain"

_antibiotics, 2022, doi:10.3390/antibiotics11040485_

Round 1
Reviewer 1 Report
The authors Belen rivaya et al presented a very good study in this manuscript. However there are a lot of flaws in the manuscript, which needs proper rectification by the others. Some of these are mentioned as below:
1. In line 20 correct the name of the bacteria and make it italic.
2. In line 27 the name of quinolones may please be corrected.
3. Line 64 needs rephrasing.
4. Line 145 write the full name of bacteria.
5. Add the number of male and female subjects participated in the study.
6. Add the procedures of prosequencing and collection of swabs in the methodology section.
7. My last and general comment is that yet i am not a native speaker but still needs rectification for language and grammatical mistakes.
Reviewer 2 Report
Main concerns:
- As stated by the authors one of the aim of this study was "…to evaluate pyrosequencing as an adequate in- house technique for detecting MG MLr, by comparing it to a commercial assay ….". However, the comparison between the pyrosequencing and the commercial available ResistancePlus® MG kit was done using only 100 samples, which even were not tested simultaneously. Does such number of samples may give a correct evaluation?
- In the current study, authors modified a pyrosequencing protocol, previously used for detection of macrolide resistance in Mycoplasma pneumoniae (ref. 28) by using the primers described by Jensen et al. for both PCR and pyrosequencing (ref. 8) (see L258-260). However, no data regarding sensitivity of this modified protocol is present. Was an internal control used, if so which one? The number of amplification cycles was increased up to 45, which may cause false positive results. Please comment. More data regarding this protocol should be presented.
- Presentation of the study design and especially the numbers, analyzed for MLr, are confusing and difficult to follow up (see L24-26: "A total of 415 MG-positive samples by Allplex STI-7 (Seegene) were analyzed for MLr detection by pyrosequencing. A total of 179 samples were further analyzed for MG and MLr by ResistancePlus® MG kit (SpeeDx) and 100 of them for fluoroquinolone resistance (FQr) by sequencing the parC gene. " and then L114-115: "….only by pyrosequencing assay (N=236 in Fig2 appeared n=215 ), only by ResistancePlus® MG kit (N=79) or tested by both methods (N=100).")
- How were 100 samples, analyzed by the both methods chosen? Were they from the prospective part of the study?
Minor corrections:
L46: make it clearly by addition of "sexually transmitted infection (STI)"
L54: use STI instead of "sexually transmitted infection (STI)"
In L240-241 written "During the second phase, a one-year prospective study was carried on, and a total of 181 samples were collected…", but Table 2 is titled "Epidemiological data and clinical findings of MG infections of 136 patients included in the prospective phase of the study." Which number is correct?
L240: 2018?
L243-249: refer to Tables 1 and 2.
L251-256 or in the results: could authors mention how much samples have been tested by Allplex STI-7 Assay (Seegene Inc., Seoul, South Korea) results in 415 MG positives?
Table 2: presentation of macrolide resistance data is not clear. For example, group of women (n=80) possesses 25/80 MLr samples identified by pyrosequencing and in the both rows, titled "Resistance detected by ResistancePlus®" and "No resistance detected (ResistancePlus®)", written "-" . Does it mean that no test was performed by ResistancePlus®?
L282: " …. This study provides valuable information about M. genitalium macrolide and fluoroquinolone resistance rates in young population with STI risk in Spain…" , change to in Badalona, Spain.
Reviewer 3 Report
Manuscript: “Detection and prevalence of macrolide and fluoroquinolone resistance in Mycoplasma genitalium in Badalona, Spain” describes comparison of pyrosequencing and one-step commercial assay in assession of the prevalence of MLr and FQr in MG in Badalona, Spain. Also the detection of MG is compared between two commercial kits. Resistance profiling is important for better treatment outcomes, in order to minimize resistance development and monitoring resistance levels in human pathogens. Thus, the work described is relevant. There are however some suggestions I want to make. In general, I would recommend to go through the entire manuscript's English language and maybe even use some professional language check service. Since I am not a native speaker I cannot specifically point out all but for example row 86: exposed à presented
Regarding the abstract: The sentence: “The comparison between Allplex and Re-sistancePlus® showed an overall agreement of 87% for MG detection that rises to 95.4% for MLr study.” is not very clear. Please specify so it can be understood without reading the entire article.
Also this sentence should be reformulated: ”Prevalence of MLr was 23.1%, but this rate rose up to 73.7% in HIV-positive patients.“ rose up is not good way to put it in my opinion.
I would change this: Despite MLr remains à remained
In figure 1 it would be good to mention that the numbers before the percentages refer to the patient amount. Also, in the text it would be good to describe if this is describtion of MG positive individuals and what subset of patients.
Please specify in table 2 MSW and MSM.
Row 132 starts more specific explanation for what has actually been compared in the study. This would maybe be easier to understand if mentioned more clear already in introduction.
Rows 208 to 210: The levels detected in the referred studies could be mentioned here
Same comment for rows 212-214.
Why not all the samples are included in the tested by both methods category?
All in all this study provides important information on the prevalence of MLr and FQr in MG in Spain. It also compares the different methodologies in detection. In my opinion larger sample size would have improved the comparison but I understand the limitations of sample collection etc. All in all good manuscript and I would recommend publication after minor revisions.
Round 2
Reviewer 1 Report
Thanks for addressing the suggestions
Reviewer 2 Report
Accept in present form